# Zinc and Its Antioxidant Properties: The Potential Use of Blood Zinc Levels as a Marker of Cancer Risk in BRCA1 Mutation Carriers

**DOI:** 10.3390/antiox13050609

**Published:** 2024-05-16

**Authors:** Milena Matuszczak, Adam Kiljańczyk, Wojciech Marciniak, Róża Derkacz, Klaudia Stempa, Piotr Baszuk, Marta Bryśkiewicz, Ping Sun, Angela Cheriyan, Cezary Cybulski, Tadeusz Dębniak, Jacek Gronwald, Tomasz Huzarski, Marcin R. Lener, Anna Jakubowska, Marek Szwiec, Małgorzata Stawicka-Niełacna, Dariusz Godlewski, Artur Prusaczyk, Andrzej Jasiewicz, Tomasz Kluz, Joanna Tomiczek-Szwiec, Ewa Kilar-Kobierzycka, Monika Siołek, Rafał Wiśniowski, Renata Posmyk, Joanna Jarkiewicz-Tretyn, Rodney J. Scott, Steven A. Narod, Jan Lubiński

**Affiliations:** 1Department of Genetics and Pathology, International Hereditary Cancer Center, Pomeranian Medical University, ul. Unii Lubelskiej 1, 71-252 Szczecin, Poland; milena.matuszczak@pum.edu.pl (M.M.); adam.kiljanczyk@pum.edu.pl (A.K.); klaudia.stempa@pum.edu.pl (K.S.); marta.bryskiewicz@pum.edu.pl (M.B.); cezary.cybulski@pum.edu.pl (C.C.); tadeusz.debniak@pum.edu.pl (T.D.); jacek.gronwald@pum.edu.pl (J.G.); tomasz.huzarski@pum.edu.pl (T.H.);; 2Read-Gene, Grzepnica, ul. Alabastrowa 8, 72-003 Dobra, Poland; wojciech.marciniak@read-gene.com (W.M.); roza.derkacz@read-gene.com (R.D.); 3Women’s College Research Institute, Women’s College Hospital, University of Toronto, Toronto, ON M5G 1N8, Canada; ping.sun@wchospital.ca (P.S.); angela.cheriyan@wchospital.ca (A.C.);; 4Department of Clinical Genetics and Pathology, University of Zielona Góra, ul. Zyty 28, 65-046 Zielona Góra, Poland; 5Department of Surgery and Oncology, University of Zielona Góra, Zyty 28, 65-046 Zielona Góra, Poland; 6OPEN, Kazimierza Wielkiego 24 St., 61-863 Poznań, Poland; godlewski.open@wp.pl; 7Medical and Diagnostic Center, 08-110 Siedlce, Poland; 8Genetic Counseling Center, Subcarpatian Oncological Hospital, 18 Bielawskiego St., 36-200 Brzozów, Poland; ajasiewicz@yahoo.com; 9Department of Gynecology, Gynecology Oncology and Obstetrics, Institute of Medical Sciences, Medical College, Rzeszow University, Rejtana 16c, 35-959 Rzeszow, Poland; 10Department of Histology, Department of Biology and Genetics, Faculty of Medicine, University of Opole, 45-040 Opole, Poland; tomiczek.onk@gmail.com; 11Department of Oncology, District Specialist Hospital, Leśna 27-29 St., 58-100 Świdnica, Poland; ewakilar@post.pl; 12Holycross Cancer Center, Artwińskiego 3 St., 25-734 Kielce, Poland; monika.siolek@wp.pl; 13Regional Oncology Hospital, Wyzwolenia 18 St., 43-300 Bielsko Biała, Poland; wiraf@poczta.onet.pl; 14Department of Clinical Genetics, Medical University of Bialystok, 15-089 Białystok, Poland; rposmyk@gmail.com; 15Non-Public Health Care Centre, Cancer Genetics Laboratory, 87-100 Toruń, Poland; 16Medical Genetics, Hunter Medical Research Institute, Priority Research Centre for Cancer Research, Innovation and Translation, School of Biomedical Sciences and Pharmacy, Faculty of Health and Medicine, University of Newcastle, Pathology North, John Hunter Hospital, King and Auckland Streets, Newcastle, NSW 2300, Australia; rodney.scott@newcastle.edu.au

**Keywords:** BRCA 1, cancerogenesis, breast cancer, ovarian cancer, cancer risk, prospective study

## Abstract

BRCA1 mutations predispose women to breast and ovarian cancer. The anticancer effect of zinc is typically linked to its antioxidant abilities and protecting cells against oxidative stress. Zinc regulates key processes in cancer development, including DNA repair, gene expression, and apoptosis. We took a blood sample from 989 female BRCA1 mutation carriers who were initially unaffected by cancer and followed them for a mean of 7.5 years thereafter. There were 172 incident cases of cancer, including 121 cases of breast cancer, 29 cases of ovarian cancers, and 22 cancers at other sites. A zinc level in the lowest tertile was associated with a modestly higher risk of ovarian cancer compared to women with zinc levels in the upper two tertiles (HR = 1.65; 95% CI 0.80 to 3.44; *p* = 0.18), but this was not significant. Among those women with zinc levels in the lowest tertile, the 10-year cumulative risk of ovarian cancer was 6.1%. Among those in the top two tertiles of zinc level, the ten-year cumulative risk of ovarian cancer was 4.7%. There was no significant association between zinc level and breast cancer risk. Our preliminary study does not support an association between serum zinc level and cancer risk in BRCA1 mutation carriers.

## 1. Introduction

In 2023, it was predicted that there would be 297,790 new cases of breast cancer in women and 19,710 ovarian cancers [1]. About 3% of breast cancers (about 7500–8500 women per year) and 10% of ovarian cancers (about 2000 women per year) are cases with BRCA1 mutations.

Approximately 13% of women in the general population will develop breast cancer during their lifetime [2]. However, in women who have inherited a deleterious BRCA1 variant, the mutation in the BRCA1 gene, the lifetime risks are 70% and 40%, respectively [2,3]. In addition to prophylactic surgery, modifiers of risks include age; hormone treatment; reproductive history; and diet, including micronutrients. Because of their extremely high risk of developing breast and ovarian cancer, we aim to find possible ways to reduce this risk.

Zinc is classified as an essential trace element and plays a crucial role in numerous cancer-suppressive mechanisms, including DNA replication, damage repair, oxidative stress response, cell cycle progression, and apoptosis [4].

Zinc functions as a cofactor for over 900 transcription factors and 300 enzymes, influencing DNA regulation, gene expression, nucleic acid synthesis, and genome stability [5]. As part of the CuZnSOD enzyme and the metallothionein protein, zinc acts as a key defender against ROS attacks [6,7,8,9]. Zinc deficiency is linked to the generation of single-strand breaks of DNA and affects repair ability, impacting processes such as repair, chromatin structure, replication, transcription, and counteracting oxidative DNA damage [10,11,12]. Moreover, zinc deficiency compromises immune responses, potentially contributing to cancer development [13,14].

There have been 18 published prospective studies on the correlation between zinc and cancer risk [5,15,16,17,18,19,20,21,22,23,24,25,26,27,28,29,30,31]. Additionally, numerous retrospective publications demonstrate a correlation between zinc and cancer risk [32,33,34,35,36,37,38,39,40,41]. To date, the role of zinc in tumorigenesis in women with BRCA1 mutations has not been studied, and for this reason, this was the purpose of our work.

## 2. Materials and Methods

The study subjects included 989 adult women, who received genetic counselling and testing between 2011 and 2017 at the Clinical Hospitals of Pomeranian Medical University in Szczecin, Poland, or at an affiliated hospital or outpatient clinic. At the first study visit, a fasting blood sample was collected from each study participant to be used for genetic testing for *BRCA1* mutations. For analysis, 10 mL of peripheral blood was collected into a vacutainer tube containing ethylenediaminetetraacetic acid (EDTA) from all study participants. All blood samples were collected between 8 a.m. and 2 p.m. and stored at −80 °C until analysis. Participants were included in the study if a deleterious *BRCA1* variant was detected.

Typically, these patients are offered the opportunity to participate in other clinical research studies. Medical charts were reviewed for date of diagnosis, age at enrollment (<50/≥50), preventive salpingo-oophorectomy (yes/no), smoking status (ever/never), oral contraceptive use (ever/never), diabetes (yes/no), dietary supplements (ever/never), hormonal therapy (ever/never), and BMI (low/normal/fat/obesity).

The study was conducted in accordance with the Helsinki Declaration and with the consent of the Ethics Committee of Pomeranian Medical University in Szczecin under the number KB-0012/73/10 of 21 June 2010. All participants provided written informed consent.

### 2.1. Measurement of Blood Zinc Level

Collected blood samples were thawed from −80 °C to room temperature on the day of analysis. Each sample was thoroughly mixed using a shaker or vortex to make the material as homogeneous as possible. This process was repeated immediately prior to taking blood volumes for dilutions due to the phenomenon of blood stratification. Using the simplest possible technique, the blood samples were diluted at a ratio of 1:30 (50 µL blood: 1450 µL buffer).

In order to achieve the specificity of the measurement, tetramethylammonium hydroxide (TMAH) solution was used for dilutions. The alkaline pH ensures good solubility of blood components, thus not causing precipitation of any of the fractions.

In addition, in order to better disperse the dissolved blood components, a non-ionic surfactant in the form of Triton X-100 was added. The use of this compound not only facilitates the dissolution of proteins, among others but also contributes to the faster flushing of the sample from the spectrometer introduction system. An internal standard in the form of rhodium (105Rh) was used to correct the matrix effect and camera drift. To achieve the stability of metal ions dissolved in solution, EDTA was used. In addition, due to the content of carbon-containing compounds, butanol was used.

The inductively coupled plasma excitation mass spectrometry (ICP-MS) technique was used to determine the content of Pb. An ELAN DRC-e mass spectrometer (PerkinElmer, Norfolk, VA, USA) and a NexION 350D mass spectrometer (PerkinElmer) were applied. Oxygen was used as a reaction gas. The ICP-MS allows for detection limits of <0.1 µg/L.

The following reference materials were used to validate the measurements: ClinCheck (Recipe, Munich, Germany), NIST 955c (National Institute of Standards and Technology, Gaithersburg, MD, USA), and BCR 634/BCR635 (European Commission, Community Bureau of Reference, Brussels, Belgium). These are reference standards commonly used in spectrometry to confirm the precision, sensitivity, and specificity of the measurement.

### 2.2. Statistical Analysis

All study participants were assigned to one of three categories (tertiles) depending on their blood zinc level. The cumulative risks of breast and ovarian cancer were calculated from the age at blood draw to the age of diagnosis of breast or ovarian cancer, death from another cause, or last follow-up. For estimating the risk of ovarian cancer, women with oophorectomy prior to blood draw were excluded, and subjects with oophorectomy in the follow-up period were censored at the time of oophorectomy. To estimate the ten-year cumulative risk of ovarian cancer, patients were followed from blood draw to date of preventive oophorectomy, ovarian cancer, ten years of follow-up, last follow-up, or death from another cause. For the analysis of breast cancer risk, oophorectomy was included as a time-dependent variable. In order to estimate the hazard ratios (HRs) for cancer risk, univariable and multivariable Cox proportional hazards regression analyses were performed. In multivariable models, the following variables were taken into analysis: zinc level (tertile), year of birth, age at blood draw (<40 years, 40–49.9 years ≥50 years), oral contraceptive use (yes/no), hormone replacement therapy use (yes/no), smoking history (current, former never), and BMI (<23.0 versus >23.0). All statistical analyses were performed using SAS, version 9.4.

## 3. Results

The study group consisted of 989 women diagnosed with a *BRCA1* mutation. The patients were followed up for an average of 6.75 years, during which time 174 new cancers were reported (121 cases of breast cancer, 29 cases of ovarian cancer, and 22 cancers at other sites). The characteristics of the study group are presented in Table 1.

The distribution of zinc levels in the cohort is presented in Figure 1.

### 3.1. Breast Cancer

There was no statistically significant correlation between blood zinc levels and breast cancer risk in *BRCA1* carriers (Table 2). For women with zinc levels in the lowest tertile, the hazard ratio was 0.88 (95% CI 0.60 to 1.29; *p* = 0.51) compared to those with zinc levels in the top two tertiles.

The distribution of zinc levels in breast cancer cases is presented in Figure 2.

### 3.2. Ovarian Cancer

Initially, unaffected women with a blood zinc level below 5797 µg/L had an increased risk of ovarian cancer, compared to women with a blood zinc level greater than 5797 (tertile 1 versus tertiles 2/3; adjusted HR = 1.95 95% CI 0.92 to 4.14), but this was not significant (*p* = 0.08). Among those women with zinc levels in the lowest tertile, the 10-year cumulative risk of ovarian cancer was 6.1%. Among those with zinc levels in the top two tertiles, the 10-year cumulative risk of ovarian cancer was 4.7% (Table 3).

The distribution of zinc levels in ovarian cases is presented in Figure 3.

### 3.3. All Cancers

Among all the 989 women, 174 developed cancers in the follow-up period. Overall, those women with zinc levels in the bottom tertile had a modestly increased risk of any cancer, compared to those in the top two tertiles (HR = 1.10; 95% CI 0.80 to 1.52). If we exclude breast or ovarian cancer, women with zinc levels in the bottom tertile had a similar risk of cancer, compared to those in the top two tertiles (HR = 1.00; 95% CI 0.70 to 1.42). There were too few cancers at other sites to provide site-specific hazard ratios for these.

## 4. Discussion

Our results demonstrated a modest and non-significant association between a low blood zinc level and an increased risk of ovarian cancer in unaffected *BRCA1* carriers. Initially, unaffected women with blood zinc levels > 5797 µg/L exhibited a twofold reduction in the risk of ovarian cancer compared to women with blood zinc levels ≤ 5797 (HR = 0.51 95% CI: 0.24–1.09), although this did not reach statistical significance (*p* = 0.08). There was no association between zinc levels and breast cancer or other cancers.

Zinc serves as a critical cofactor for enzymatic activities such as dehydrogenases, peptidases, and zinc finger domains. Zinc is involved in a number of reactions necessary for the proper functioning of the human body (Table 4).

The recommended daily value of zinc is 11 mg for men and 8 mg per day for women. Thus far, there has been no suggested blood zinc level; however, the recommended concentration of zinc in serum or plasma typically ranges from 800 to 1200 mcg/L.

Zinc can be absorbed through several pathways, including passive diffusion and absorption in the digestive tract, regulated by transporters [48]. The bioavailability of zinc in the digestive tract increases in the presence of citric acid and decreases in the presence of iron, calcium, phosphorus, fiber, and phytate [49]. Individuals with a vegetable-rich diet may exhibit lower zinc absorption rates. For example, legumes contain a relatively high amount of zinc (Table 5), but the presence of phytate, which inhibits the absorption of zinc, results in less of this element being supplied to the body than in the case of providing the same amount from animal foods [50].

There have been 18 published prospective studies on the correlation between cancer risk and zinc [5,15,16,17,18,19,20,21,22,23,24,25,26,27,28,29,30,31]. There are numerous publications that demonstrate a correlation between zinc and cancer risk [32,33,34,35,36,37,38,39,40,41]. However, these are retrospective papers, and for this reason, they were not analyzed further in our publication.

We found 18 prospective studies about zinc and cancer risk (Table 6). Of these, there were eight papers on colorectal, five on prostate, two on breast, two on pancreatic, one on hepatocellular, one on lung, and one on kidney cancer. Among them, 13 showed a positive correlation between low zinc levels in the blood and cancer risk, but the remaining 7 did not show a statistically significant result. In most studies, the exposure data were based on questionnaire information about intake. The exception is one prospective study [15], in which zinc levels were measured in serum and urine.

In another study [15], in addition to the association with zinc and copper levels, the strongest correlation was shown between the highest quartile Cu/Zn ratio in serum and urine (OR, 2.37; 95% CI, 1.32–4.25). Even for serum alone, the ratio was better than for each micronutrient separately (OR—1.75; 95% CI: 1.21–2.54). Elevated copper and low zinc levels are the most common trace metal imbalances encountered in the human body [51].

Zinc interacts with the human body through a variety of mechanisms, which are crucial for its proper functioning. This is, for example, evidenced by the fact that metalloprotease activity mediates every stage from (ovarian) tumor formation to metastatic implantation [52].

The results of this study have several potential clinical implications. If confirmed, the evaluation of zinc levels and the levels of other microelements in the blood of BRCA1 carriers may be used as a marker of the presence of early cancers and as a risk factor for later cancer development. This information is potentially relevant for BRCA1 mutation carriers who are considering preventive oophorectomies. Notably, our study revealed that around 33% of women demonstrated low zinc values and would be candidates for supplementation. In the future, blood testing and dietary advice and/or supplement use might be used to optimize zinc levels among BRCA1 carriers.

In summary, our study did not prove that blood zinc levels are associated with the risk of cancers among BRCA1 carriers. However, there was a suggestive association between low zinc levels and a higher risk of ovarian cancers. It is important to perform further investigations and observations on a larger number of carriers and with longer follow-ups.

## 5. Conclusions

In summary, our preliminary study does not confirm an association between serum zinc levels and cancer risk in BRCA1 mutation carriers. We hypothesize that zinc levels may predict lower risks of ovarian cancer, but the correlation was not statistically significant. Further studies are needed. Additionally, there is a need to generate results with women with other genetic mutations. 

## Figures and Tables

**Figure 1 antioxidants-13-00609-f001:**
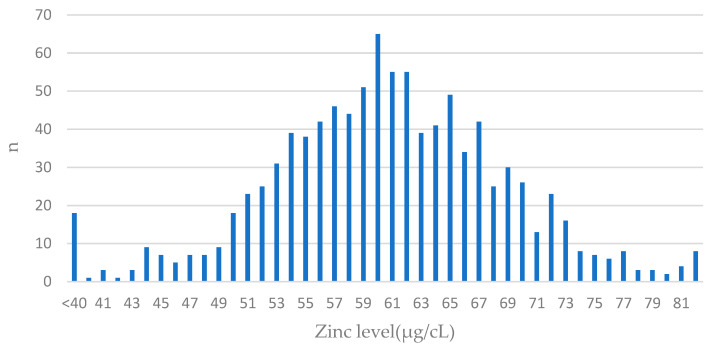
The distribution of values of zinc levels in blood among BRCA1 carriers. Features of normal distribution can be seen. The largest number of patients had blood levels close to the mean value (61 µg/cL) in the entire group; n—number of patients.

**Figure 2 antioxidants-13-00609-f002:**
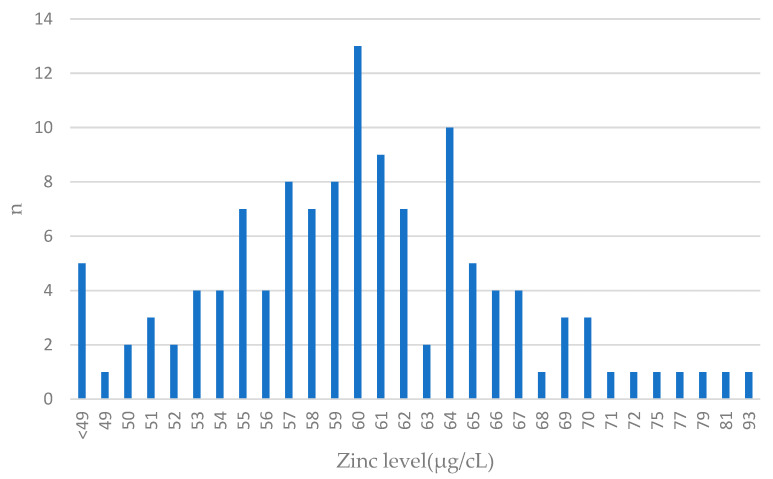
Zinc levels in blood among breast cancer cases. Features close to normal distribution can be seen. The largest number of patients had blood levels close to the mean value (61 µg/cL) in the entire group; n—number of patients.

**Figure 3 antioxidants-13-00609-f003:**
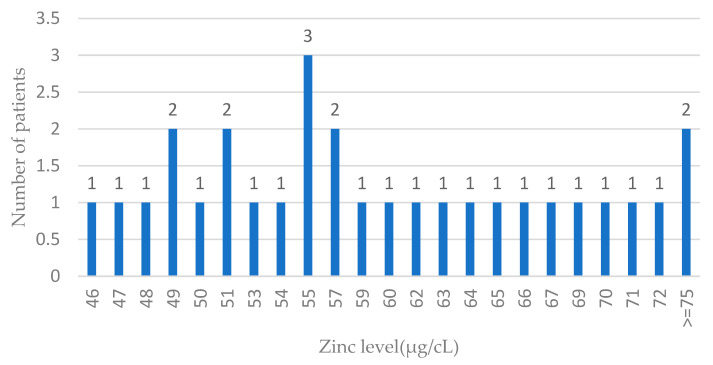
Zinc levels in blood among ovarian cancer cases. Features of the normal distribution cannot be seen (probably due to the low number of ovarian cases); n—number of patients.

**Table 1 antioxidants-13-00609-t001:** Group characteristics.

	N = 989
Age at enrollment	
<50 years	775 (78.36%)
≥50 years	214 (21.64%)
Smoking	
never	720 (72.80%)
ever	264 (26.69%)
missing data	5 (0.51%)
Hormonal therapy	
never	720 (72.80%)
ever	263 (26.59%)
missing data	6 (0.61%)
Oophorectomy	
no	413 (41.76%)
yes	576 (58.24%)
missing data	0 (0.00%)
Oral Contraceptive use	
never	501 (50.66%)
ever	481 (48.64%)
missing data	7 (0.70%)
Diabetes	
no	880 (88.98%)
yes	62 (6.27%)
missing data	47 (4.75%)
Body Mass Index (kg/m^2^)	
<18.5	56 (5.66%)
18.5–24.9	553 (55.92%)
25.0–29.9	237 (23.96%)
≥30.0	95 (9.61%)
missing data	48 (4.85%)
Dietary supplements usage	
never	500 (50.56%)
ever	489 (49.44%)
New cancer site (n = 174) (by the first cancer)	
breast	122 (70.11%)
ovarian	29 (16.67%)
bladder	2 (1.15%)
cervix	3 (1.72%)
colon	2 (1.15%)
kidney	1 (0.57%)
leukemia	2 (1.15%)
lung	3 (1.72%)
pancreas	1 (0.57%)
salivary gland	1 (0.57%)
sarcoma	1 (0.57%)
site unknown	1 (0.57%)
skin	1 (0.57%)
thyroid	3 (1.72%)
urothelial	1 (0.57%)
abdomen–CSU	1 (0.57%)

**Table 2 antioxidants-13-00609-t002:** The hazard ratio for breast cancer according to zinc level (tertiles).

Variables	Breast Cases/ Total	Univariate HR (95% CI) P	Multivariate * HR (95% CI) P
Zinc µg/L			
≤5797	38/329	1	1
5797–6433	51/329	1.38 (0.91–2.10) 0.13	1.39 (0.90–2.12) 0.13
>6433	34/331	0.90 (0.57–1.44) 0.67	0.91 (0.57–1.47) 0.70
Total	123/989		
Zinc			
<5797	38/329	1	1
>5797	85/660	1.14 (0.78–1.67) 0.51	1.15 (0.78–1.70) 0.48
Year of birth			
≤1965	38/239	1	1
January 1965–1975	28/224	0.79 (0.49–1.29) 0.35	0.61 (0.26–1.41) 0.25
January 1975–1985	43/337	0.85 (0.55–1.31) 0.45	0.66 (0.20–2.16) 0.49
1985	14/189	0.58 (0.31–1.07) 0.08	0.40 (0.11–1.44) 0.16
Age at blood draw (years).			
≤40	62/566	1	1
40.01–50	30/216	1.22 (0.79–1.90) 0.36	1.55 (0.65–3.73) 0.32
>50	31/207	1.28 (0.83–1.79) 0.26	1.18 (0.36–3.90) 0.78
Oophorectomy			
No	30/413	1	1
Yes (time-dependent)	93/576	0.87 (0.61–1.26) 0.46	0.64 (0.39–1.03) 0.07
Oral contraceptive use			
No	59/502	1	1
Yes	64/481	1.10 (0.78–1.57) 0.58	1.22 (0.83–1.78) 0.32
HRT			
No	91/720	1	1
Yes	32/263	0.82 (0.55–1.23) 0.34	0.78 (0.50–1.78) 0.32
Smoking			
No	59/553	1	1
Current	35/222	1.52 (1.00–2.31) 0.05	1.51 (0.99–2.30) 0.06
Former	29/209	1.30 (0.83–2.03) 0.25	1.22 (0.78–1.92) 0.38
BMI at blood taken			
≤median (23.05)	63/464	1	1
>median (23.05)	57/477	0.84 (0.59–1.21) 0.35	0.77 (0.53–1.14) 0.19
Missing	3/48		

* Adjusted by all the variables listed in the left column.

**Table 3 antioxidants-13-00609-t003:** Hazard ratios (HRs) for ovarian cancer by zinc level (tertiles).

Variables	Ovarian Cases/ Total	Univariate HR (95% CI) P	Multivariate * HR (95% CI) P
Zinc µg/L			
≤5797	13/259	1	1
5797–6432	6/261	0.45 (0.17–1.19) 0.11	0.40 (0.15–1.07) 0.07
>6433	10/262	0.76 (0.33–1.74) 0.52	0.63 (0.27–1.47) 0.28
Total	29/782		
Zn ≤ 5797	13/259	1	1
Zn > 5797	16/523	0.61 (0.29–1.26) 0.18	0.51 (0.24–1.09) 0.08
Year of birth			
≤1965	10/101	1	1
January 1965–1975	9/164	0.49 (0.20–1.22) 0.13	0.94 (0.07–12.1) 0.96
January 1975–1985	9/328	0.25 (0.10–0.64) 0.003	0.28 (0.02–4.82) 0.38
>1985	1/189	0.06 (0.01–0.50) 0.006	0.05 (0.00–1.59) 0.09
Age at blood (years)			
≤40	14/556	1	1
40.01–50	5/129	1.53 (0.55–4.23) 0.42	0.46 (0.12–1.72) 0.25
>50	10/97	4.49 (1.99–10.1) 0.0003	1.10 (0.07–18.0) 0.95
Oral contraceptive use			
No	18/374	1	1
Yes	11/402	0.54 (0.25–1.14) 0.10	0.79 (0.35–1.83) 0.59
HRT			
No	26/662	1	1
Yes	3/154	0.40 (0.12–1.32) 0.13	0.33 (0.10–1.10) 0.07
Smoking			
Never	12/447	1	1
Current	7/176	1.46 (0.58–3.71) 0.42	1.40 (0.55–3.60) 0.48
Former	10/154	2.53 (1.09–5.85) 0.03	2.23 (0.93–5.32) 0.07
BMI at blood draw			
≤23	11/396	1	1
>23	16/339	1.70 (0.79–3.65) 0.18	1.06 (0.45–2.49) 0.90
Missing	2/47		

* Adjusted by all the variables listed in the left column.

**Table 4 antioxidants-13-00609-t004:** The effect of zinc on carcinogenesis.

Essential Component	Zinc Is Crucial for More than 900 Transcription Factors (i.e., ZF DNA-Binding Domains), 300 Enzymes (i.e., CuZnSOD; DNA Repair Proteins) [42]
Enzymatic functions, impact on DNA, gene expression, nucleic acid synthesis, and genome stability	Zinc plays a key role in the activity of many enzymes, including those involved in DNA repair and control of cell growth. Zinc regulates gene expression via ZF transcription factors (DNA repair genes) [43]. Zinc deficiency is associated with the generation of single-strand breaks on DNA and affects repair ability [10]. Both BER and NER systems contain ZFP and other zinc-related proteins [11]. Moreover, affects DNA by being a part of the repair process, chromatin structure, replication, and transcription [11,12] and acts against oxidation DNA damage [10].
Apoptosis	Caspases are activated by Zn; they are involved in the process of apoptosis.
	Zn affects a number of signaling pathways (involved in apoptosis), i.e., p53 and heat shock pathways.
	Zn modulates the ratio between Bcl-2 family proteins and by them regulates apoptosis.
Detoxification	Metallothionein, a protein binding diverse metal ions (e.g., Cd, Pb, Zn, and Cu), helps regulate metal ion levels in cells, forming stable complexes that aid in eliminating these metals from the body, thus protecting against harmful effects [44]. Additionally, it neutralizes ROS by contributing to cell protection from oxidative stress and preventing damage [45].
Immune response	The lack of this element may compromise immune responses, potentially contributing to cancer development. It plays a role in the cytolytic activity of T lymphocytes [13].
Antioxidant function	As part of the CuZnSOD enzyme, it acts as a key defender against ROS attacks. It serves as an antagonist to redox-active transition metals, such as Fe and Cu, preventing the oxidation of sulfhydryl groups in proteins. This protective role extends to sulfhydryl-containing proteins like tubulin and ZFP, as well as alanyl tRNA synthetase, guarding against thiol oxidation and disulfide formation and providing protection against free radicals [6,7,8,9].
Regulation of signaling pathways	Zn^2+^ regulates signaling pathways in both directions through among others p38 and regulation of histone acetylation and ZFP. Zinc-deficient cells are unable to maintain normal p53 expression [46,47].
Regulation of inflammation	Zn^2+^ inhibits inflammation through suppression of Nf-kB [14].

BER—base excision repair; Cd—cadmium; CuZnSOD—copper–zinc superoxide dismutase; NER—nucleotide excision repair, Pb—lead, ROS—reactive oxygen species, Fe—iron, ZF—zinc finger ZFP—zinc finger protein; Zn—zinc.

**Table 5 antioxidants-13-00609-t005:** The average content of zinc and DV in selected foods with favorable bioabsorption.

Food	Zinc Content Per 100 g	Daily Value
Shellfish (Oysters)	39.3 mg	300–413%
1. Alaska King Crab 2. Shrimp, mussels	1. 7.62 mg 2. 1.6 mg	1. 69–95% 2. 15–20%
1. Nuts (i.e., almonds). Seeds 2. Sunflower 3. Hemp	1. 5.78 mg 2. 5.29 mg 3. 4.34 mg	36–63%
1. Red meat (beef) 2. Offals 3. Poultry (chicken breast)	1. 4.79 mg 2. 1.7 mg 3. 0.68 mg	1. 38–55% 2. 13–15% 3. 5%
1. Cheese 2. Eggs 3. Milk (1 cup)	1. 3.74 mg 2. and 3. 1 mg	1. 30–40% 2. and 3. 5–13%
Fish (1. Salmon 2. Flounder/sole)	1. 0.5 mg 2. 0.32 mg	3–4%

**Table 6 antioxidants-13-00609-t006:** Prospective studies on cancer risk.

Neo.	Cohort	Follow-Up (Years)	Results	Other Relevant Findings	Ref.
Lu	Cases (n = 440) Control (n = 1320)	4	Elevated plasma zinc levels were linked to a decreased risk of lung cancer OR = 0.89 (95% CI: 0.79–0.99)	Better results were achieved in men [OR = 0.86; 95% CI = 0.74; 0.99]. Zinc levels in individuals who developed cancer had lower plasma zinc levels compared to the healthy cohort (1183.13 vs. 1275.48 *p* = 0.019).	[22]
Br	Cases (n = 1186) Control (n = 1186)	19	No significant associations were detected between zinc levels, whether measured in serum or obtained from dietary sources prior to diagnosis, and the risk of breast cancer. The adjusted odds ratio (OR) for breast cancer in serum zinc quartile 4 (Q4) compared to quartile 1 (Q1) was 1.09 (95% CI: 0.85–1.41), while for zinc intake, the OR for Q4 versus Q1 was 0.97 (95% CI: 0.77–1.23).	Furthermore, no distinct associations were observed between zinc and any characteristics of breast cancer. The kappa value of 0.025 (*p* = 0.022) indicated poor agreement between serum zinc and zinc intake.	[19]
Br	Cases (n = 496) Control (n = 496)	2	High levels of Zn were associated with a reduced risk of breast cancer OR = 0.56 (95% CI: 0.33–0.95; *p* = 0.031) for women with zinc levels in the highest tertile in both plasma and urine compared with the lowest. The risk remained consistent regardless of the ER/PR/HER2 status.		[15]
HCC	Cases (n = 106) Control (n = 106)	6.5	In the case of hepatocellular carcinoma (HCC), there was a strong inverse relationship observed between the highest and lowest tertiles for zinc levels (OR = 0.36; 95% CI: 0.13–0.98, *p* = 0.0123).	The calculated Cu/Zn ratio demonstrated a positive correlation with HCC (OR = 4.63; 95% CI: 1.41–15.27, *p* = 0.0135). Furthermore, each 20 μg/dl increase in circulating zinc was associated with a 45% reduction in HCC risk (OR = 0.55; 95% CI: 0.39–0.78) in model 1 and a 47% reduction (OR = 0.53; 95% CI: 0.33–0.84) in model 2.	[17]
CRC	Cases: W (n = 498) M (n= 789) Control: W (n = 44,878) M (n = 38,932)	5	The quartile of men with the highest zinc intake had (HR = 0.77; 95% CI: 0.57–0.85) reduced risk of CRC among men.	However, in multivariate-adjusted models, zinc intake was not significantly associated with CRC risk among men; the coefficients for the highest quartile versus the lowest quartile of zinc intake were HR = 0.77 95% CI: 0.58–1.03 for colorectal cancer, HR = 0.76 95% CI: 0.54–1.07 for colon cancer and HR = 0.80 95% CI: 0.49–1.32 for rectal cancer. In women, there was no significant association between zinc intake and CRC risk in any of the models.	[23]
CRC	Cases: W (n = 192) M (n = 344) Controls W (n = 72,593) M (n = 59,636)	W 15.2 M 9.3	No statistically significant correlation between dietary intake of zinc and the risk of colon cancer.	The results showed a trend toward lower zinc values (mg/day) in cancer cases compared to controls, but these results were not statistically significant (10.4 ± 1.0 vs. 10.5 ± 1.1 for women and 12.2 ± 1.3 vs. 12.3 ± 1.4 for men).	[18]
CRC	Cases (n = 966) Controls (n = 966)	9.11	An inverse association with cancer risk was observed for higher levels of zinc (OR = 0.65; 95% CI: 0.43–0.97; *p* = 0.07).	Copper was also statistically significant, and consequently, the copper–zinc ratio was positively associated with CRC (OR = 1.70; 95% CI: 1.20–2.40; *p* = 0.0005).	[5]
CRC	Controls (n = 34,708) Cases—proximal (n = 438) Cases—distal (n = 303)	15	High dietary zinc intake may decrease the risk of colon cancer (proximal and distal). Multivariable RR= 0.38 (CI: 0.17–0.74; *p* = 0.01) compared referent quartile vs. the highest intake for proximal colon cancer. Zinc intake was also associated with a decreased risk of distal colon cancer (RR = 0.55; CI: 0.30–1.02; *p* = 0.03).	The inverse association with zinc intake was stronger among women who consumed alcohol than among those who did not.	[24]
CRC	Cases W (n = 1079) M (n = 1035) Cohort W (n = 121,700) M (n = 51,529)	22	In comparing the highest quartile (Q4) with the lowest quartile (Q1) of dietary zinc intake, the multivariable relative risks (RRs) were 0.86 (95% CI: 0.73–1.02) for colorectal cancer, 0.92 (95% CI: 0.76–1.11) for colon cancer, and 0.68 (95% CI: 0.47–0.99) for rectal cancer. The notable inverse association observed between dietary zinc intake and the risk of rectal cancer was predominantly influenced by data from women, although the difference in the sex-specific results did not reach statistical significance.		[25]
CRC	Cases (n = 990) Cohort (n = 54,208)	13	There were no significant results for high and low zinc intake among smokers and CRC (HR = 1.38; 95% CI: 1.14–1.68; *p* = 0.28). There were also no results for non-smokers, and the effect was even less significant (HR = 1.10; 95% CI: 0.9–1.35).	A statistically significant association was observed between a low overall intake of vitamin E, selenium, manganese, and zinc, as well as the never use of only selenium and zinc supplements, and a more than 14% increased risk of CRC compared to those with a high intake.	[26]
Pr	Cases (*n* = 6980) Controls (n = 47,240)	28.3	Men who used zinc supplements for 15 years or more had an elevated risk of fatal prostate cancer (HR: 1.91, 95% CI: 1.28–2.85, *p* < 0.001) and aggressive prostate cancer (HR: 1.55, 95% CI: 1.03–2.33, *p* = 0.004).	Moreover, in comparison to individuals who never used zinc supplements, men who consumed more than 75 mg/day of supplemental zinc exhibited an increased risk for lethal prostate cancer (HR: 1.76, 95% CI: 1.16–2.66, *p* = 0.001) and aggressive prostate cancer (HR: 1.80, 95% CI: 1.19–2.73, *p* = 0.006).	[27]
Pr	Cases (n = 1706) Controls (n = 2404)	5 Gr.: 1. ≤1 2. 1–4 3. 5–9 4. 10–14 5. ≥15	Using zinc supplements for ten years or longer was associated with a more than twofold increase in the risk of advanced prostate cancer (RR = 2.3, 95% CI: 1.1–5.0) compared with no zinc use.	The authors concluded that zinc has an adverse effect of zinc on prostate cancer carcinogenesis (OR = 1.9; 95% CI: 1.0–3.6; *p* = 0.6–0.8). In these analyses, the (OR = 2.1, 95% CI: 1.1–4.1) for using zinc for ≥10 remained significantly elevated.	[28]
Pr	Cases (n = 2901, of them advanced 434) Controls (n = 46,974)	14	The use of zinc supplements for a duration of 10 years or more in men was associated with a relative risk of 2.37 (95% CI = 1.42–3.95; *p* < 0.001).	The results showed that excessive high-dose supplementation >100 mg/day increased the risk of advanced breast cancer RR = 2.29 (95% CI = 1.42–3.95; *p* < 0.001). This study demonstrated that prolonged intake of excessive amounts of zinc supplements may lead to elevated carcinogenic processes.	[29]
Pr	Cases (n = 832, of them advanced 123) Controls (n = 34,410)	3.5	An average 10-year intake of zinc supplementation > 15 mg/day, when compared to non-supplementation, did not show a statistically significant association with an overall reduced risk of prostate cancer (HR = 0.82; 95% CI: 0.58–1.14; *p* = 0.44).	Adequate zinc supplementation (>15 mg/day for 10 years) was linked to a decreased risk of advanced prostate cancer (either locally invasive or with distant metastasis, n = 123) compared to no supplementation (HR = 0.34; 95% CI = 0.13–1.09; *p* = 0.04). Dietary zinc, however, did not show an association with prostate cancer.	[30]
Pr	Cases (n = 392) Controls (n = 783)	5	There was no indication supporting a reverse correlation between serum zinc levels and the risk of prostate cancer. The average serum zinc concentrations showed no significant difference between cases (94.9 µg/dL) and controls (93.9 µg/dL) (*p* = 0.42). Moreover, no association was observed between serum zinc levels and prostate cancer, either in the overall analysis or when considering tumor stage/grade.	However, the authors noted a hint in the results specific to ethnicity suggesting a potential rise in risk. In ethnicity-specific analyses, positive associations were identified in Japanese Americans (odds ratio for the highest vs. the lowest tertile = 2.59, 95% CI: 1.09–6.17) and Latinos (odds ratio = 2.74, 95% CI: 1.05–7.10), while no association was observed in African Americans and whites.	[16]
Ren	Cases (n = 229) Controls (n = 63,028)	20.1	Dietary zinc was found to be positively correlated with kidney cancer risk; the highest quartile relative to the lowest (Q1 vs. Q4 HR = 1.74; 95% CI: 1.02–2.97; *p* = 0.033).		[31]
Pan	Cases (n = 49) Controls (n = 3970)	10	There were inverse non-significant correlations in this case between the lowest quartile and the sum of the three higher for zinc (HR = 0.91, 95% CI 0.44 to 1.91, *p* = 0.81).		[20]
Pan	Cases (n = 184) Controls (n = 77,446)	7.1	No association was observed between zinc use and the incidence of pancreatic cancer (Q1 vs. Q3 HR = 0.94; 95% CI: 0.52–1.71; *p* = 0.98).		[21]

Br—breast cancer; CRC—colorectal cancer; Gr—groups; HCC—hepatocellular cancer; M—men; Neo—malignant neoplasm; Lu—lung; Pan—pancreas; Pr—prostate; Ref.—reference; Ren—kidney; W—women.

## Data Availability

Data supporting the results presented are available from the authors upon request from any interested researchers.

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
