# Peer review of "Zinc and Its Antioxidant Properties: The Potential Use of Blood Zinc Levels as a Marker of Cancer Risk in BRCA1 Mutation Carriers"

_antioxidants, 2024, doi:10.3390/antiox13050609_

Round 1

Reviewer 1 Report

The introduction must be correct, the material is too little. I would insert the topic of table 6 which is in the discussion.

In the case study I would also have placed controls (without BRCA1) to see if zinc varied with the same characteristics (smokers, oral contraceptives, etc.).

Line 128: In table 1 they have to align the numbers correctly.

Author Response

Dear reviewer, 
thank you for your suggestions. They are very enriching and contribute to improving the substantive quality of our work. Your recommendations have been taken into account and used in the revisions. Please have a look at the attached new version of the paper with the tracker of changes. Thank you for your help in improving the manuscript and please feel free to make further suggestions. 

Thank you for your collaboration and best regards,

Jan

Reviewer 2 Report

 I strongly recommend that the authors re-statistically analyze blood zinc level(Figure 1)between each cancer risk in the BRCA1 group.  The conclusions is too simple for the clinical research to publish. Page 12, line 223:  The abstract is not compatible with the conclusions. Authors should rewrite the conclusions. 

Manuscript ID: antioxidants-2954894

Title: Zinc and its antioxidant properties. The potential use of blood zinc levels as a marker of cancer risk in BRCA1 mutation carriers.

The manuscript “Zinc and its antioxidant properties. The potential use of blood zinc levels as a marker of cancer risk in BRCA1 mutation carriers.” by Milena Matuszczak et. alis a good clinical research in a current and interesting topic and could be of interest for pharmaceutical and medical scientists.

In this study, the authors analyzed the zinc level from 989 female BRCA1 mutation blood sample to find the risk of ovarian cancer. The manuscript is very well written, the methodology and results clearly described and discussed. The results reported are of interest for the scientific community. This research may contribute to the pharmaceutical industry and cancer therapy. There are a couple minor issues that would affect the publication of this research article in its current form.

1.          Authors should revise a couple typing errors.

2.          Page 7, line 169-170: Authors should correct the following chemical formula and transcription factor protein (Zn2+ through suppression of NF-κB).

3.          I strongly recommend that the authors re-statistically analyze blood zinc level(Figure 1)between 

        each  cancer risk in the BRCA1 group.

4.          Page 12, line 223: The conclusions is too simple for the clinical research to publish Authors should 

         rewrite the conclusions.

5.          The abstract is not compatible with the conclusions.

Author Response

Dear Reviewer,

I would like to express my sincere gratitude for the insightful suggestions you provided regarding our manuscript. Your input has proven invaluable in enhancing the substantive quality of our work. I am pleased to inform you that we have diligently incorporated your recommendations into the revised version of the paper.

Attached herewith, you will find the updated manuscript along with a change tracker for your convenience. Your continued assistance in improving the manuscript is highly appreciated, and we welcome any further suggestions you may have.

Thank you once again for your invaluable contribution to our research endeavor.

Best regards,

Prof. Jan Lubinski
